# RANDOM SCALING OF EMERGENT CAPABILITIES

## ABSTRACT

Language models famously improve under a smooth scaling law, but some specific capabilities exhibit sudden breakthroughs in performance. While advocates of "emergence" view breakthroughs as unlocked capabilities, others attribute them to thresholding effects on noncontinuous metrics. We propose that breakthroughs are instead driven by continuous changes in the *probability distribution* of training outcomes when performance is bimodally distributed across random seeds. In synthetic length generalization tasks, we show that different random seeds can produce either highly linear or emergent scaling trends. We reveal that sharp breakthroughs in metrics are produced by underlying continuous changes in their distribution across seeds. In a case study of inverse scaling, we show that even as the probability of a successful run declines, the average performance of a successful run increases monotonically. We validate our distributional scaling framework on realistic settings by measuring MMLU performance in LM populations. Our observations hold true even under continuous loss metrics, confirming that random variation must be considered when predicting a model's performance from its scale.

## 1 INTRODUCTION

On most benchmarks, language model (LM) performance is determined by a scaling law (Hestness et al., 2017; Rosenfeld et al., 2019; Kaplan et al., 2020) that varies smoothly with parameter size and overall training compute. There are, however, a number of celebrated exceptions in which performance abruptly improves on specific benchmarks (Srivastava et al., 2023). These sudden breakthroughs fuel one of the most heated debates in modern AI.

On one side, advocates of **emergence** claim that performance abruptly improves at particular scales because those scales provide the capacity to learn specific concepts (Wei et al., 2022). On the other side, skeptics argue that these sudden improvements are a *mirage* driven by thresholding effects and alleviated by more appropriate continuous metrics—though a few **breakthrough capabilities** remain stubbornly emergent (Schaeffer et al., 2024). We will argue that such discontinuities are driven by continuous changes in the *probability* of a breakthrough at each scale. In other words, the discontinuities are real—each model firmly either *knows* or *does not know* a given concept—but emergent breakthroughs do not always reflect some fixed threshold which permits the concept to be learned. Instead, models may learn the concept at various scales, albeit with changing probability.

We posit that a breakthrough capability is distinguished not by deterministic responses to scale, but by *multimodal* random variation. In other words, independent training runs cluster in their performance metrics. This observation is undocumented because scaling laws usually plot a single training run at each scale, rarely testing multiple seeds. Although random variation may be benign when model performance is measured in-distribution (Jordan, 2024), previous work suggests that out-of-distribution performance may vary widely across training runs (Zhou et al., 2024a;b; Qin et al., 2024; Juneja et al., 2023; Li et al., 2025), even at larger scales (Madaan et al., 2024).

By connecting breakthrough scaling with random variation, we challenge the narratives of both the emergence and mirage camps. First, our results *contest the position of the mirage or "loss-to-downstream" camp*, which claims that effects of scale are predictable and that continuous metrics will smooth out apparently discrete concept learning. We discover clustered multimodal distributions of capabilities, confirming that models unpredictably learn critical discrete concepts—and these concepts are observable even using continuous loss metrics. Furthermore, we *complicate the narrative of the competing emergence camp*, which often treats specific model sizes as distinct in their capacity.

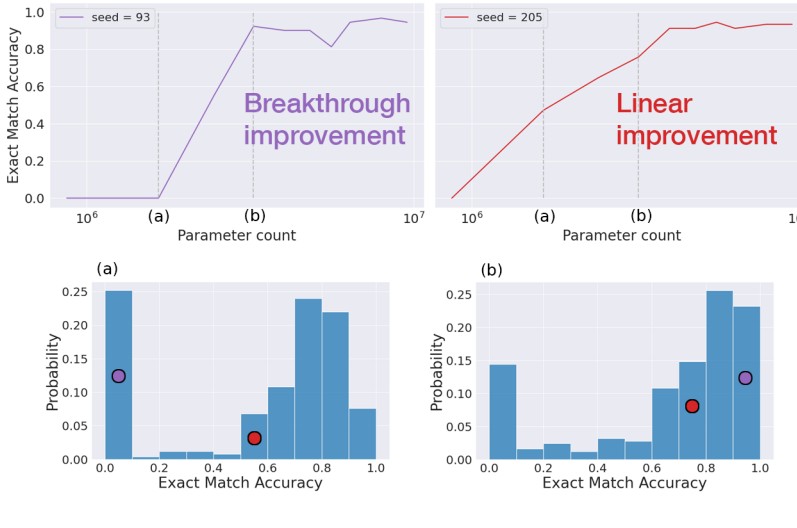

Figure 1: **Different random seeds produce different scaling trends.** Scaling trends can be emergent or linear for different seeds, even if all models train on the same data with the same hyperparameters. On the count task, we show trends for random seeds with the highest breakthroughness (seed 93; top left) and linearity (seed 205; top right). We mark parameter counts immediately before and after seed 93's emergence respectively as (a) and (b). Histograms illustrate the bimodal distribution of performance across all random seeds at scales (a) and (b), marking the positions of seeds 93 and 205. Breakthroughs occur when consecutive points represent different clusters; linear trends occur when each point is sampled from the same gradually shifting cluster.

While we confirm that algorithms require some minimum model capacity, that capacity may not be the enormous scale at which emergence is observed. Instead, *discontinuous* performance jumps are sampled from a *continuously* changing multimodal distribution, where the "success" component ultimately dominates at larger scales. To extrapolate downstream metrics to large scales, we must predict the likelihood of successful emergence as well as the performance of a successful run.

Since training numerous seeds is prohibitively expensive at large scales, we study partially reinitialized LLMs and toy models. Whereas prior work reports summary statistics across only a few training runs, we characterize the full multimodal performance distribution. We find:

- **Breakthroughs result from bimodal performance distributions.** In multiple choice (Section 3.1) and synthetic task length generalization (Section 2.1), some random seeds produce linear scale trends while others are emergent. This variation is caused by the bimodal distribution of each skill across random seeds (Section 2.2), a property that materializes around **breakthrough thresholds** in model size. At these scales, emergence is a stochastic property (Section 2.3).
- **Bimodal variation persists under continuous metrics.** In the emergence debate, one main position (Schaeffer et al., 2024) is that breakthroughs are caused by measuring discontinuous metrics such as accuracy rather than loss. For our tasks, continuous loss metrics can remain visibly bimodally distributed, particularly when there is a more even split of failed and successful runs. We confirm these loss distributions on both synthetic (Section 2.4) and natural (Section 3.4) tasks.
- **When a scale curve exhibits sudden *discontinuous* improvement in a skill, the *probability of learning that skill may be changing continuously*.** Treating the bimodal distribution as a mix of **failure** and **success** distributions, we illustrate that average improvements can come from changes in the probability of success *or* in the mean performance of a successful run (Section 2.2). Although bimodality can appear abruptly at a minimum capacity scale (Section 2.5), these sudden distributional changes do not necessarily align with breakthrough scales for individual model runs.
- **In realistic settings, multimodal performance clusters emerge based on data composition and model scale.** In Section 3, we extend our analysis to the task of multiple choice question answering by training partially reinitialized LMs. Our findings confirm that random variation in LMs is also bimodal for established emergent capabilities near their breakthrough threshold.

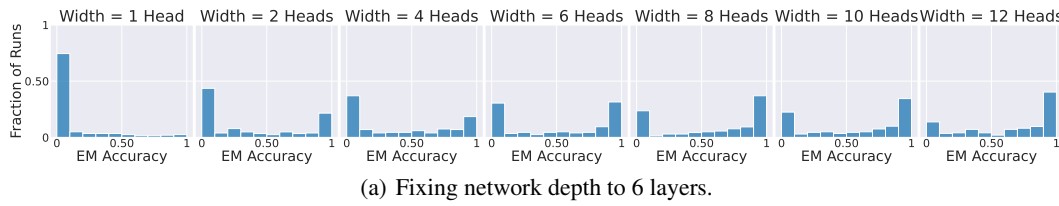

(a) Fixing network depth to 6 layers.

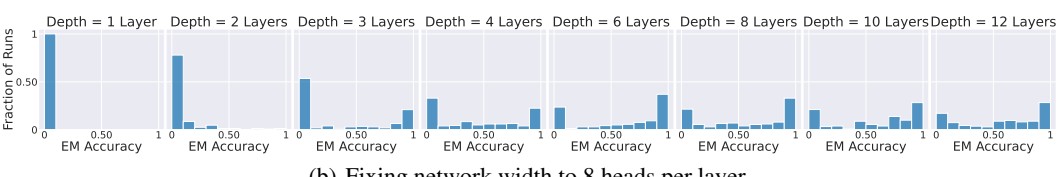

(b) Fixing network width to 8 heads per layer.

Figure 2: **Random variation in length generalization (addition task).** Histograms of exact match accuracy on length 40 sequences when independently scaling (a) width and (b) depth.

## 2 SYNTHETIC LENGTH GENERALIZATION EXPERIMENTS

Usually, performance breakthroughs are measured on a single model per scale or, at most, the average of a few runs. The literature suggests that these emergent capabilities unlock at specific model scales (Wei et al., 2022), implying that scaling curves for different seeds would perform similarly to within a margin of noise. Contrary to this belief, we demonstrate that performance is sampled from a stochastic distribution which changes gradually even as individual scaling curves jump abruptly.

### 2.1 TASKS AND SETUP

After training models on synthetic tasks, we measure their length generalization (Graves et al., 2016; Kaiser and Sutskever, 2015; Lake and Baroni, 2018; Hupkes et al., 2020), one of many compositional properties that can lead to conceptual breakthroughs (Srivastava et al., 2023; Löwe et al., 2024; Chen et al., 2024). Further experimental details are in Appendix A.

**Architecture:** In our synthetic experiments, we train decoder-only Transformer models from scratch using rotary position embeddings (RoPE) (Su et al., 2024). To observe the random performance distribution at each scale, we train our models from hundreds of seeds. We choose model sizes by separately adjusting either the width (number of 64-parameter heads per layer) or depth hyperparameter.

**Task:** We consider two algorithmic tasks previously studied in Zhou et al. (2024a): counting and reverse order addition. Performance on these task ranges widely across random seeds. At each scale, we train models from 250 seeds for the count task and 200 seeds for reverse order addition.

- **Count**: Given two numbers in increasing order, the model is trained to generate a sequence which counts consecutively from the first number to the second number. Examples are given in the form `"5, 9 >, 5, 6, 7, 8, 9"`, while limiting the length of the counting sequence during training. Zhou et al. (2024a) showed that models trained to count can generalize to more than twice this training length; however, we will reveal a more nuanced view of length generalization based on its distribution across independent model runs.
- **Addition**: Zhou et al. (2024b) showed that Transformers can generalize 10-15 digits past training length for an addition task, if provided with index hints and allowed to generate the answer backwards. Examples take the form `"a0, 3, a1, 4, +, a0, 2, a1, 8, >, a1, 2, a0, 6"`, where index hints are a consecutive sequence of the max number of digits sampled randomly from 0 to the max evaluation length. We use this modified form of reverse order addition with index hints as our addition task.

**Dataset:** During training, we sample sequences i.i.d from the train set and invoke in-context learning by adding examples to the context, following prior work (Jelassi et al., 2024; Zhou et al., 2024b). The lengths of examples are sampled uniformly from 1 to the maximum training length (30 for count and 35 for addition). Length generalization is then tested at length 60 for count and 40 for addition.

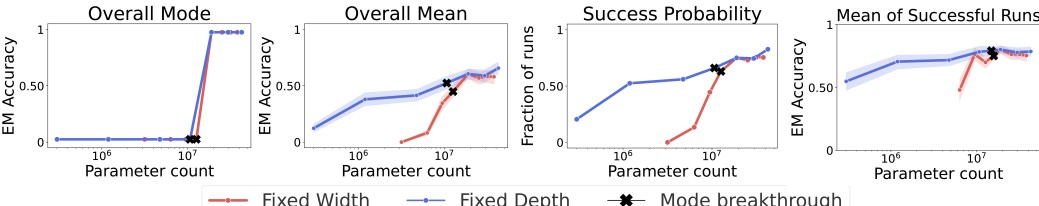

Figure 3: **Summary statistics for length generalization (addition task).** Exact match statistics for 200 models trained at length 35 and tested at length 40. We track overall mode and mean in the two leftmost plots. Because the EM accuracy distribution is bimodal, the mode exhibits a sharp increase even as the mean evolves continuously. Defining success as $> 20\%$ EM accuracy, we also note the continuous change in the probability of success and in the mean of successful runs in the two rightmost plots. Means feature 95% confidence intervals over 1000 bootstrapped samples.

## 2.2 EMERGENCE IS A SIGN OF BIMODAL VARIATION

What do the scaling curves for length generalization tasks look like when we generate them in conventional ways? Following Srivastava et al. (2023), we take the vector of model performances for the count task at test length 60 across different scales with a fixed initialization and shuffle seed.[1] We calculate the task's *breakthroughness* and *linearity* metrics on each seed. As defined in Appendix B, the former metric measures emergence, whereas the latter measures a smooth response to scale. We plot the performance across scale for the seeds with the highest breakthrough and highest linearity in Figure 1. As seen in Figure 1 and Appendix Figure 9, we can easily find fixed seeds that show varying levels of emergence and linearity, due to random variation in breakthroughs.

This variation is explained by the bimodality of model performance distributions when varying seed. Figure 2 illustrates that, for a population of models independently trained on the addition task, length generalization clusters into high and low component modes at many parameter sizes. This clustering produces distinctly bimodal performance distributions, causing some model runs to appear as breakthroughs while others follow a more linear progression. When bimodally distributed runs cluster into very high and very low performance components, a model might exhibit linear scaling if sampled from the same cluster as the previous scale *or* emergent scaling when switching from the low cluster to the high cluster. These differences ultimately lead to high variability in the timing and degree of emergence. Furthermore, these differences can be a source of oscillating scale curves, as Srivastava et al. (2023) documented in their lowest-linearity tasks.

The subsequent sections further probe the connection between the bimodal nature of the performance distribution and emergence of length generalization capabilities. We focus on addition here with results on the count task in Appendix C.1. Interestingly, the counting task exhibits periods of *inverse scaling* in average improvement, a phenomenon also observed in LMs (McKenzie et al., 2022) which we discuss in more detail in the Appendix.

## 2.3 SUDDEN JUMPS FROM GRADUAL DISTRIBUTION SHIFTS

When reporting metrics from only one seed or the mean of a few seeds, the outcome is likely to be close to the *mode* performance of the underlying model population. As shown in Figure 3 ("Overall Mode"), the mode of the performance distribution shows a massive spike in improvement mirroring an emergent benchmark's scaling trend. However, we claim that this discontinuous improvement is only an artifact of sampling the *mode* performance, although the underlying bimodality also produces discontinuity in many single-seed scaling curves. Underlying this *discontinuous* performance jump are *continuous* changes in other distributional statistics. In Figure 3 ("Overall Mean"), the mean exhibits a smoother trend in accuracy. Mode and mean diverge thus because the underlying distribution is bimodal, expressing a mixture of "successful" and "failing" runs.

---

[1]Although it is standard practice to fix the random seed when reporting LM benchmark performance across scales, the initializations produced by a single seed have no meaningful relation across different scales.

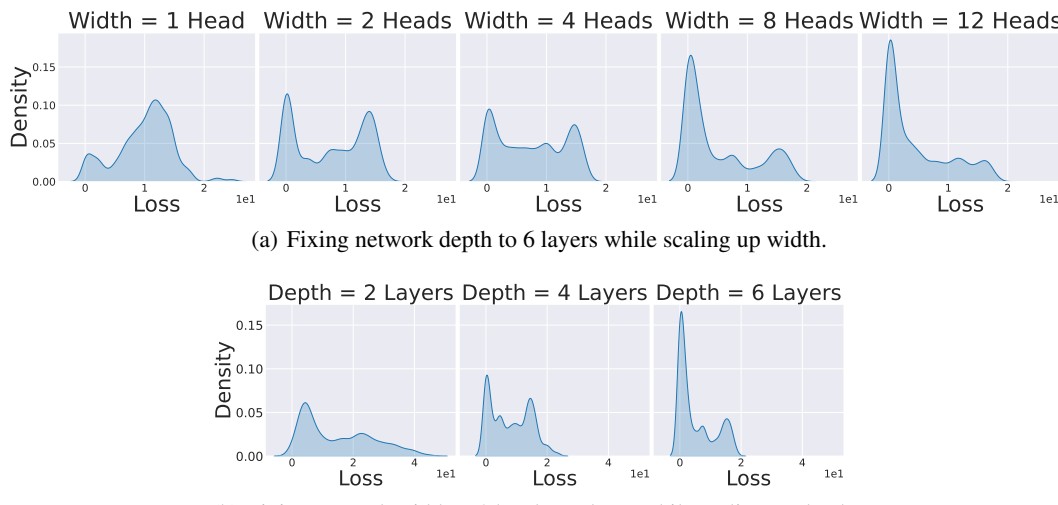

(a) Fixing network depth to 6 layers while scaling up width.

(b) Fixing network width to 8 heads per layer while scaling up depth.

Figure 4: **Random variation in continuous length generalization error (addition task).** Kernel Density Estimation (KDE) of our loss-based error metric (Equation 1) across model runs. At scales where the EM accuracy distribution is bimodal, the distribution remains bimodal even when using a continuous metric.

Treating the distribution as a mixture of successes and failures, we can separately analyze the *probability* of a successful run and the *performance distribution* of successful runs. Both of these properties are changing continuously and gradually when the mode increases abruptly. If we restrict our analysis to the runs achieving nontrivial 20% accuracy, we see that the probability and mean of such "successful" runs (Figure 3, two rightmost plots) both exhibit continuous improvement, with the exception of increasing from depth 2 to 3 (discussed further in Section 2.5). Even at the mode breakthrough,these underlying distributional properties are only changing gradually. We conclude that when tasks exhibit bimodal distributions across random seeds, there are statistics exhibiting continuous improvements underlying the seemingly abrupt improvements across scale.

### 2.4 Is Bimodality a Mirage?

Metrics with hard thresholds and discontinuities can artificially induce breakthrough behavior (Schaeffer et al., 2024); conversely, continuous metrics turn apparent emergence into smooth curves (Srivastava et al., 2023). We must be particularly cautious about claiming emergence when requiring outputs to exactly match a target string, as we have so far. Are our case studies artifacts of thresholding effects? Do our bimodal distributions become unimodal under continuous metrics?

To avoid thresholding artifacts, we use a continuous equivalent to the exact match metric: the maximum loss assigned to any individual token. Each token is individually computed in the correct output context, so the entire sequence represents a fixed set of continuous per-token loss functions. Because the maximum of a fixed set of Lipschitz-continuous functions is always Lipschitz-continuous, this error score is guaranteed to be continuous as long as per-token loss is continuous. For model $f$ on dataset $X$, the continuous error score based on per-token loss $L(f(x_{0\ldots i-1}), x_i)$ is:

$$\text{error}(f, X) = \frac{1}{|X|} \sum_{x \in X} \max_{i < |x|} L(f(x_{0\ldots i-1}), x_i) \tag{1}$$

In Figure 4, we plot this continuous metric for addition models on the length generalization test dataset. The distribution of this metric across random seeds is often still clustered. We therefore confirm that emergent capabilities exhibit bimodal performance distributions even when using a continuous performance metric; their bimodality is not due to thresholding alone. Further discussion around continuous metric plots for count and histograms for probability-based metrics are in Appendix C.3.

## 2.5 BIMODALITY EMERGES ABRUPTLY

The previous section showed that bimodal performance distributions produce emergent scaling curves for *individual seeds*. We next analyze how the *distribution itself* evolves across scales. As shown in Figure 2, the performance distribution starts as unimodal at the smallest scale, where no models can length generalize. Larger scales yield a bimodal distribution where most of the probability mass is ultimately placed on successful length generalizing runs. A priori, there could be a smooth evolution between these two distributions in which probability mass from failing runs gradually shifts towards higher performance metrics, eventually splitting into a clear separate cluster. We find instead that the shift from unimodal to bimodal is abrupt and instantly polarized into low and high clusters. In Appendix Figure 13 we show that models fail to generalize when fixing depth to be one layer even when scaling width (despite having perfect in-distribution accuracy).

To quantify this abrupt transition, we track how the performance distribution changes across scales. Specifically, we plot the Wasserstein-L2 distance of each distribution relative to that of the final, largest model scale when separately fixing width and depth.

In Figure 5 we see that there is a sharp decrease in the W2 distance for scaling with a fixed width, quantifying the sudden appearance of highly successful runs when model depth reaches 3 layers or model width reaches 2 heads. These sudden changes identify the moment when a new capability is unlocked, as the distribution transitions abruptly from unimodal on one extreme to bimodal at both extremes. We posit that this transition marks the **minimum capacity** required to learn the task. We also mark the point in each trend where the mode in Figure 3 increases sharply. We posit that it may be misleading to draw conclusions about minimal model capacity on a specific task using single runs at each scale, whereas distributional metrics correctly predict the sudden appearance of probability mass placed on successful runs.

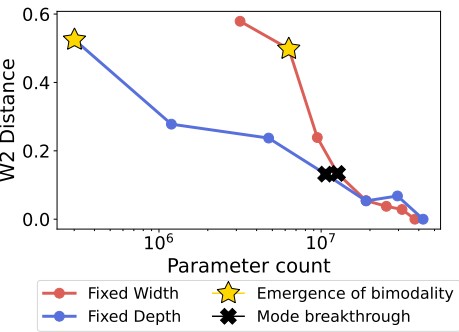

Figure 5: **Changes in random variation (addition task).** Wasserstein-L2 distance of each scale's performance distribution relative to the largest scale, scaling depth and width independently. The sharpest change in distance occurs early.

## 3 LLM EXPERIMENTS

After connecting emergence with bimodality in small synthetic settings, we turn to LMs. We focus on the MMLU dataset, where high performance emerges after the LM learns the multiple choice format (Srivastava et al., 2023; Hu and Frank, 2024). To avoid the expense of repeatedly training large-scale models from scratch, we simulate independent runs by reinitializing the upper layer of pretrained LMs before continuing to train them. Because Qwen base models perform well for their scale (39.5% for Qwen2.5-0.5B), these experiments effectively require models to *re*-discover MMLU capabilities after their removal. Our continued training dataset, a mix of C4 news data and the MMLU auxiliary training data, is insufficient to recover the full MMLU capability of the the base model.

### 3.1 DATA AND SETUP

We extend our investigation to LMs by testing multiple choice question answering, an emergent natural language understanding task (Srivastava et al., 2023; Snell et al., 2024). We test whether the performance clusters observed in emergent synthetic tasks also manifest in LMs. Specifically, we hypothesize that emergent scaling curves in LMs express underlying bimodal (or multimodal) performance distributions.

Our procedure can be regarded as continued pretraining: we mimic the model's original pretraining objective with a dataset of English language sequences. We hypothesize that at the emergence point for this task, performance distributions will also be bimodal across random seeds. During pretraining, a large diverse corpus encourages models to acquire various capabilities. While sufficiently large models may learn all such capabilities, smaller models have limited capacity, requiring these capabilities to

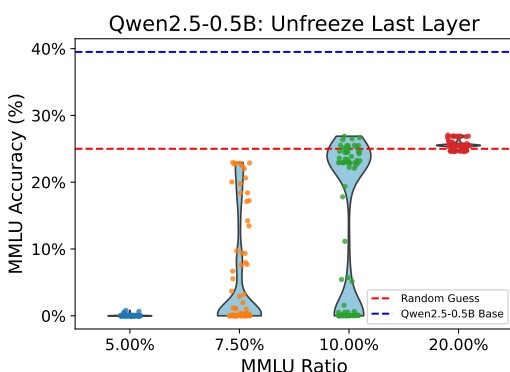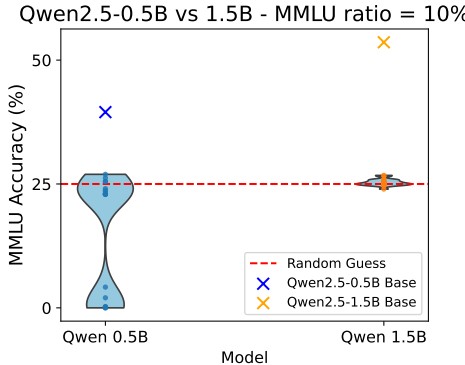

(a) Qwen2.5-0.5B trained on different mixes.  (b) Qwen2.5 models trained on 10% MMLU data.

Figure 6: **Performance on MMLU.** All models have their last layer reinitialized randomly and then are trained on a mix of C4 news and MMLU, with varying proportions of MMLU. *Left:* With insufficient MMLU training data, each model fails to follow the multiple-choice format, resulting in trivial performance. As more MMLU data is included, different seeds lead to bimodally distributed performance. With enough MMLU data, all models score above the random baseline. *Right:* For a fixed data mix, the smaller model has bimodally distributed performance. In contrast, the larger model consistently relearns to answer multiple-choice questions, though not at its original base performance.

compete (Merrill et al., 2023). This competition, influenced by initialization and data order, leads to varying outcomes across random seeds, forming performance clusters for specific benchmarks.

**Task**: We test LMs on MMLU (Hendrycks et al., 2021), a multiple-choice question-answering benchmark. Strong performance on MMLU requires two key abilities: (1) natural language reasoning with domain-specific knowledge and (2) producing correct answers in the required format. It is this latter ability which leads to emergent trends (Srivastava et al., 2023; Hu and Frank, 2024).

**Model**: We use the Qwen2.5 family of base models (Yang et al., 2024). To introduce randomness during continued pretraining, we reinitialize the final attention layer and the subsequent LM head. We perform full-parameter continued pretraining on these reinitialized models.

**Data**: To ensure that the multiple choice formatting circuits compete with other tasks during continued pretraining, we create a diverse dataset by mixing the English news subset of C4 (Raffel et al., 2023) with the official MMLU auxiliary training data. The C4 news data reinforces general language modeling ability, while the MMLU data focuses on multiple-choice question answering. We vary the proportion of MMLU in this mixture to control the target task's data size in addition to model size, as both are common control variables in the scaling laws literature.

**Training**: We continue pretraining the Qwen2.5-0.5B and Qwen2.5-1.5B models on our C4-MMLU mixes, training 80 reinitializations on each data mixture ratio. We train Qwen2.5-0.5B models for 2 epochs and Qwen2.5-1.5B models for 5 epochs to ensure convergence on the MMLU validation set. We use a learning rate of 1e-5 with linear decay scheduler.

We focus our analysis on experiment results from Qwen2.5-0.5B. In Appendix D, we include results from Qwen2.5-1.5B (Appendix D.1) as well as a second task CommonSenseQA (Talmor et al., 2019) (Appendix D.2).

## 3.2 EMERGENCE ACROSS DATA COMPOSITIONS

We first examine how data composition influences the bimodal capabilities of Qwen2.5-0.5B. Specifically, as the training data contains more examples of a target breakthrough task, it changes the task's performance distribution in ways that mirror the effect of scale.

In Figure 6 (*left*), when trained on a data mix containing only 5% MMLU examples and 95% random sequences from C4 (Raffel et al., 2023), most models (out of 80 seeds) achieve near 0% performance on the MMLU test set. According to the existing literature, this failure stems from a failure to process the multiple-choice format (Hu and Frank, 2024). As the proportion of MMLU samples

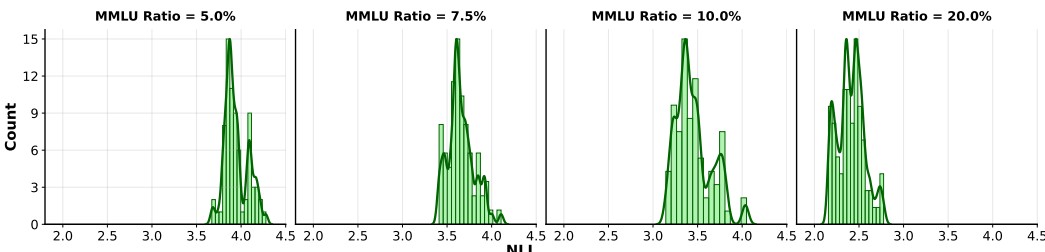

Figure 8: **NLL loss distributions on the MMLU test set.** Each distribution is provided as a histogram and as a smoothed KDE. Even with a continuous metric (NLL), we still observe multimodal distribution across random seeds.

increases, bimodality emerges when MMLU composes 10% of the training data, where models form two distinct performance clusters. In one cluster, continually pretrained models still fail to follow the correct format, while in the other, models achieve performance at or above the random guess baseline of 25%. This second cluster contains models that consistently respond with valid multiple choice options; those that outperform the random baseline have even learned to compose the multiple choice format with their world knowledge when selecting an answer. Finally, with sufficient data ($> 20\%$ MMLU), models across all 80 seeds consistently outperform the 25% random baseline.

**Quantifying multimodality.** To verify the distributional pattern in Fig. 6, we use *Hartigan's Dip Test* Hartigan and Hartigan (1985). The test fits the closest unimodal cumulative distribution function (CDF) to the empirical CDF of model accuracies across seeds and measures the largest vertical gap between them, referred to as the *Dip* score. By definition, $\text{Dip} \in (0, 0.25]$: values near zero indicate a shape that closely follows a unimodal distribution, while larger values reflect stronger deviation from unimodality. We then

Table 1: Hartigan's Dip Test on MMLU accuracies across 80 seeds at different training data ratios. Larger *Dip* implies greater deviation from unimodality.

| MMLU Ratio | Dip ↑ | $p$-value |
|---|---|---|
| 5.00% | 0.047 | 0.368 |
| 7.50% | 0.057 | 0.110 |
| 10.00% | 0.144 | < 0.001 |
| 20.00% | 0.107 | < 0.001 |

perform a statistical test using the Dip statistic and report the associated $p$-value, which quantifies whether the observed multimodality is statistically significant. A smaller $p$-value implies that the observed distribution is unlikely to have arisen from a unimodal population, providing stronger evidence for multimodality. As shown in Table 1, at 5% the $p$-value is large (consistent with unimodality); at 7.5% the $p$-value is 0.11, suggesting a mild deviation; and at 10% and 20% we obtain $p < 0.001$, indicating statistically significant multimodality in the distribution of outcomes across seeds.

### 3.3 EMERGENCE ACROSS MODEL SCALES

Like in the synthetic setting, LM model size affects the clustered performance distribution. Using a training data mixture with 10% MMLU auxiliary training samples, we continually pretrain Qwen2.5-1.5B on the same 80 seeds and report MMLU test accuracy in Figure 6 (*right*). While models built on the smaller Qwen2.5-0.5B form two distinct performance clusters, those built on the larger Qwen2.5-1.5B consistently outperform the random baseline. In fact, larger models reliably acquire MMLU capability when trained on the same dataset that produces highly bimodal variation at smaller scales. This conclusion recalls the literature on scaling laws, which argues that smaller models require more training examples to match the performance of larger models (Rosenfeld et al., 2019; Kaplan et al., 2020). Appendix D.2 shows similar results on the Commonsense QA (Talmor et al., 2019) task.

### 3.4 Is MMLU emergence a mirage?

Is multiple choice performance only bimodal because accuracy is thresholded? As in our synthetic setting (Section 2.4), we must address the effect of the discontinuous metric. In Figure 8, we find that continuous negative log likelihood (NLL) loss metrics remain highly multimodal. The 10% MMLU ratio leads to the most even split between successes and failure in accuracy, and accordingly to clear bimodality in NLL loss, mirroring the link we observed on synthetic tasks. Notably, training data with a 7.5% MMLU ratio produces a performance distribution with a small "success" cluster peaked around NLL=3.4. The 10% MMLU ratio produces a performance distribution with a larger success cluster, also peaking around NLL=3.4. Therefore, the continuous metrics remain roughly bimodal—as is especially clear in the 10% mix setting, which is most evenly split between success and failure.

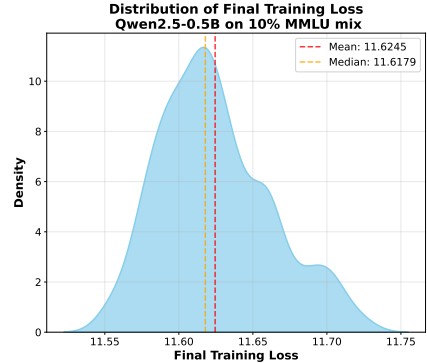

Figure 7: **KDE approximation of LM training loss on 10% MMLU mixture.** Approximation of the distribution across 80 random seeds is unimodal, smooth, and nearly symmetrical with median close to mean. By contrast, Figure 8 depicts MMLU loss for LMs trained on the same mix as irregularly clustered with multiple modes.

While recent arguments against emergence (Schaeffer et al., 2024) claim that discontinuities disappear with continuous loss metrics, Figure 8 reveals the opposite effect in random variation—NLL loss reveals clusters that are *concealed* by accuracy metrics, specifically among low-performance models like those trained on the 5% MMLU mix. (To easily compare these metrics, see Figure 14.)

Is this random clustering particular to the emergent task? As an alternative hypothesis, we might expect all tasks—including training loss—to have similarly multimodal NLL distributions, when approximated with our KDE settings. To the contrary, the pretraining loss produces a smooth, unimodal, and almost symmetric KDE distribution (Figure 7). *We therefore conclude that multimodality is linked to emergent tasks—not merely a downstream effect of in-distribution training noise.*

## 4 Discussion

Our work explores the evolution of random variation in model performance across scales, bringing a more nuanced perspective on emergent capabilities. The mode of our performance distributions sharply improves at a certain model scale, similar to emergent capabilities in the literature, but we attribute these sudden jumps to gradual improvements in the random distribution. Furthermore, we show that bimodality can emerge *before* the mode—or most seeds—exhibit a breakthrough (Section 2.5); the transition from the initial unimodal distribution to a bimodal one is sudden.

**Random variation:** Model performance can be sensitive to stochastic aspects of the training process like random initialization and training data order. Prior studies have documented performance differences across various stress test sets (D'Amour et al., 2022; Naik et al., 2018). Most relevant to us, the high variability of length generalization is well-documented (Zhou et al., 2024b;a). More generally, out-of-distribution behavior like compositional rules (McCoy et al., 2019) or associative biases (Sellam et al., 2022) often exhibit extreme variation compared to in-distribution loss. Such differences persist throughout training, not just at the final checkpoint (Zhou et al., 2020). Dodge et al. (2020) compared the impacts of weight initialization and data ordering, concluding that both contribute equally to variation in performance. Existing work has also found model runs can cluster in shortcut learning (Juneja et al., 2023) and in training dynamics (Qin et al., 2024; Hu et al., 2023), hinting at multimodal variation. Our work connects these random clustering effects to the phenomenon of emergence at scale.

**Length generalization and compositionality with transformers:** Transformer models cannot always generalize to longer sequences than they are trained on (Anil et al., 2022; Deletang et al., 2023; Gontier et al., 2020; Hupkes et al., 2020; Schwarzschild et al., 2021; Zhang et al., 2022). Various proposals have, consequently, aimed to improve length generalization. Some methods focus

on alternative positional encodings (Shaw et al., 2018; Press et al., 2022; Su et al., 2024; Kazemnejad et al., 2024; Jelassi et al., 2024). Others modify the dataset format by adding scratchpad or Chain-of-Thought (Anil et al., 2022) or by incorporating padding and index hints (Jelassi et al., 2023; Zhou et al., 2024a). Regarding random variation, Zhou et al. (2024a) and Zhou et al. (2024b) provide evidence of variability in length generalization across random seeds, which we further investigate and analyze across a range of model scales.

**Emergent abilities of LMs:** In LLMs, emergent abilities are behaviors that arise unexpectedly as models are scaled up in size or trained on larger datasets (Hestness et al., 2017; Rosenfeld et al., 2019; Brown et al., 2020; Kaplan et al., 2020). These abilities are characterized by unpredictable and abrupt performance improvements on specific benchmarks at certain scales (Wei et al., 2022; Ganguli et al., 2022; Srivastava et al., 2023). Understanding the conditions and mechanisms underlying emergence is a key area of research. Although recent studies suggest that some breakthroughs may stem from the choice of evaluation metrics rather than fundamental changes in model behavior with increased scale (Srivastava et al., 2023; Schaeffer et al., 2024), some breakthrough capabilities remain emergent. Snell et al. (2024) found that some scales exhibit earlier emergence if finetuned explicitly on an emergent task, suggesting that smaller models may have the capacity for that task but are limited by its scarcity in the training corpus. Similarly, we show that emergent capabilities can arise from multimodal random variation using synthetic length generalization tasks as a case study.

**Depth versus Width Scaling:** Although scaling laws offer a smooth extrapolation for model performance, downstream performance can vary depending on architecture shape and not just model size (Tay et al., 2022). For compositional tasks, deeper models often generalize better up to a certain point, but for a fixed compute budget, it may be more advantageous to train a shallower, wider model (Petty et al., 2024). Various works have proposed explanations for the role of width versus depth in scaling behavior; for instance, Edelman et al. (2024) show that increasing network width offers more 'parallel queries' over randomized subnetworks which learn sparse features more efficiently. Levine et al. (2020) use a border rank argument to establish a width-dependent depth threshold, beyond which additional depth yields diminishing returns. In this work, we specifically investigate how independently scaling width and depth influences the random variation distribution in compositional tasks. In one surprising result, we document a regime in which increasing width damages model performance but increasing depth improves model performance. In a research community increasingly concerned with emergent capabilities, our findings should inspire further study of how emergent tasks respond to architectural hyperparameter tradeoffs.

## LIMITATIONS

Our study is constrained by computational resources. Ideally, we would pretrain full language models from scratch across many model scales and many random seeds, but end-to-end pretraining at this density is prohibitively expensive. Instead, our LM experiments rely on partially reinitialized Qwen2.5 models at a small number of scales, which may not capture all aspects of large-scale training dynamics. While these models use a standard decoder-only Transformer architecture and thus are representative of many contemporary LMs, our conclusions about distributional scaling and bimodality should be validated on additional architectures and pretraining pipelines. Finally, although we document when and where bimodal performance distributions appear, we do not fully characterize their mechanistic origin—for example, how optimization landscapes, data composition, or competition between circuits give rise to distinct performance clusters. Understanding these causal mechanisms is an important direction for future work.

## REPRODUCIBILITY STATEMENT

Upon publication, we will release the code for both synthetic and LLM experiments, including code, and data. Detailed descriptions of model, data and training are provided in Appendix A, and Section 3.

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

## A    EXPERIMENTAL DETAILS ON SYNTHETIC TASKS

Below we provide more details when training decoder-only language models on the count and addition—with index hints—tasks. Hyperparameters are largely adapted from (Zhou et al., 2024a). We train all of our models to convergence on the train distribution. Our experiments were run on an internal cluster and all model scales can be run on a single 40GB A100 GPU with gradient accumulation; for count, runs can finish within 2 hours and for addition, runs can finish within 6 hours.

**Count task:** For all of our training runs, we fix the vocabulary size to 150. For evaluation, we compute the exact match (EM) accuracy across all consecutive subsequences of the test length.

- **Model scales:** As mentioned in Section 2.1, we scale up our models by fixing width and scaling depth and fixing depth and scaling width. The precise parameters for each variation are as follows. For our **fixed depth experiments**, we fix the network depth to 4 layers and vary width by taking hidden dimensions $\{64, 128, 256, 384, 512, 640, 768, 1024\}$. The head dimension is fixed to 64. For our **fixed width experiments**, we fix the hidden dimension to be 512 and vary the depth from $\{1, 2, 4, 6, 8\}$ layers.
- **Hyperparameters:** We use a learning rate of $1e-3$ with a cosine decay scheduler and weight decay 0.1. We set the maximum training duration to be 10000 steps, with batch size 128 and context length 256.

**Reverse Order Addition with Index Hints:** For evaluation, we compute the exact match (EM) accuracy across 500 batches of 128 examples each.

- **Model scales:** For our **fixed depth experiments**, we fix the network depth to 6 layers and vary width by taking hidden dimensions $\{64, 128, 256, 384, 512, 640, 768\}$. For our **fixed width experiments**, we fix the hidden dimension to be 512 and vary the depth from $\{1, 2, 3, 4, 6, 8, 10, 12\}$ layers.
- **Hyperparameters:** We use a learning rate of $1e-4$ with a cosine decay scheduler and weight decay 0. We set the maximum training duration to be 30000 steps, with batch size 64 and context length 512.

## B    BREAKTHROUGHNESS AND LINEARITY

Srivastava et al. (2023) introduced *breakthroughness* and *linearity* metrics to capture model performance improving suddenly or reliably with scale. Given a model's performances $y_i$ at model scales $x_i$ sorted by ascending model scale, the linearity metric $L$ and breakthroughness metric $B$ are respectively calculated as

$$L = \frac{I(y)}{\text{RootMeanSquare}(\{y_{i+1} - y_i\}_i)},$$

$$B = \frac{I(y)}{\text{RootMedianSquare}(\{y_{i+1} - y_i\}_i)}$$

where $I(y) = \text{sign}(\arg\max_i y_i - \arg\min_i y_i)(\max_i y_i - \min_i y_i)$.

In Figure 9 we sample the five top seeds for the breakthroughness and linearity metric respectively for count (above) and addition (below).

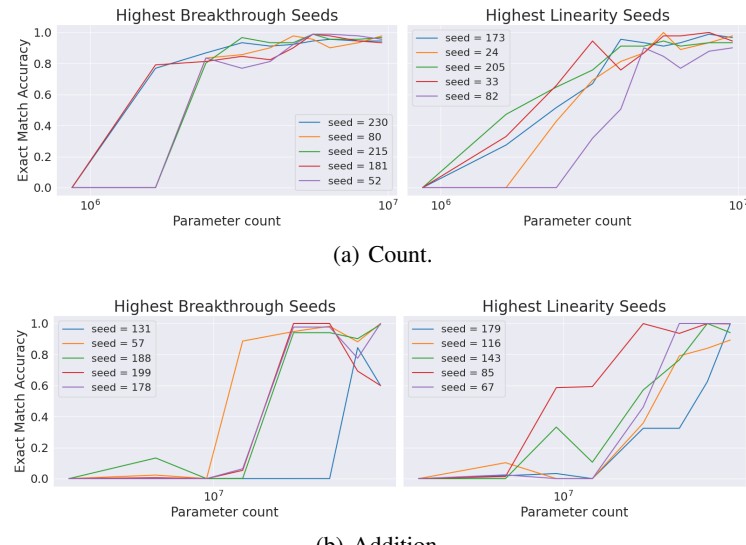

(a) Count.

(b) Addition.

Figure 9: **Top five seeds according to breakthroughness and linearity metrics.** Definitions for the two metrics are given in Appendix B, with resulting seeds plotted for (a) count and (b) addition.

## C ADDITIONAL SYNTHETIC EXPERIMENTAL RESULTS

### C.1 COUNT: AN INSTANCE OF U-SHAPED SCALING

We next consider the counting task (Section 2.1), which also exhibits bimodally-distributed performance (see Appendix Figure 12) but yields a very different scaling effect: a U-shaped curve. Figure 10 reveals this peculiar phenomenon in the mean accuracy scaling curve, when holding depth fixed. This curve is not simply a result of the summary statistic chosen, as it is mirrored by the evolution of the distributions as a whole according to their W2 distance in Figure 11.

U-shaped scaling has been observed in LMs, but its causes are not currently well-understood. When the Inverse Scaling Prize (McKenzie et al., 2022) solicited tasks which exhibit inverse scaling trends—performance decreasing with scale—for large models, Wei et al. (2023) revealed that the majority of awarded tasks actually exhibit U-shaped scaling after considering even larger models. Treating the counting task as a concrete instance of U-shaped scaling at small model scales, we find that this unusual trend is still underscored by monotonic continuous changes in the performance distribution. Indeed, Figure 10 (c) shows that although the trend in the mean across all runs is U-shaped curve, the mean of the "successful" runs—those achieving at least 50% accuracy—still improves monotonically when increasing width. The observation of inverse scaling is, instead, due to changes in the *probability* of success (bottom left). Even when inverse scaling is in effect across a performance distribution, the performance of successful runs may exhibit more conventional responses to scale.

### C.2 ADDITIONAL EXACT MATCH ACCURACY HISTOGRAM PLOTS

Our plots in the main paper fixed test length 40 for addition (Figure 2). In Figure 12 we show analogous histograms for a fixed test length of 60 on the count task. We note that unlike addition, the mode corresponding to failing runs disappears at the highest widths and depths ran, indicating a difference in task difficulty affecting the random variation distribution.

In Section 2.5, we posit that the performance distribution becomes bimodal at the **minimum capacity** required by the task. For example, Figure 13 shows that, regardless of width, models with a fixed depth of 1 layer are unable to length generalize on the count task, despite achieving near-perfect accuracy in-distribution.

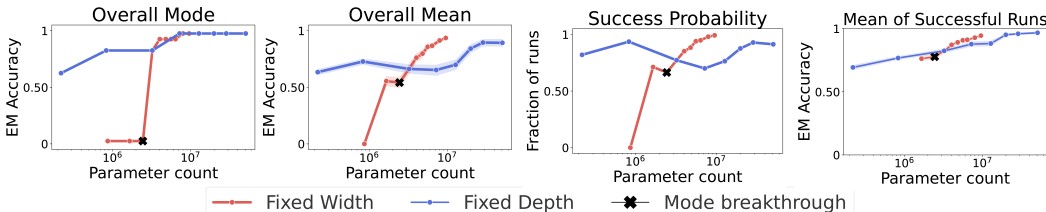

Figure 10: **Summary statistics for length generalization (count task).** Models tested at length 60, scaling width (with fixed depth 4) and depth (with fixed width 4) independently. The trend for fixed depth (scaling width) exhibits a U-shaped curve for the mean across *all* runs (b); however, the mean of *successful* runs (defined as >50% EM accuracy) (d) still improves continuously.

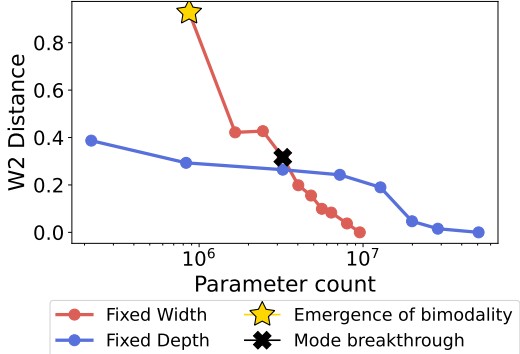

Figure 11: **Changes in random variation (count task).** Wasserstein-L2 distance of each scale's performance distribution relative to the largest scale, scaling depth and width independently. The distribution changes slowly between intermediate model scales, but changes suddenly at the nadir of the U-shaped curve in Figure 10. We mark the emergence of bimodality at the last scale before multiple peaks appear, and the mode breakthrough at the last scale before successful length generalization becomes marginally more likely than failure.

## C.3 ADDITIONAL CONTINUOUS METRIC PLOTS

In Section 2.4 we have seen seen that the addition histograms are bimodal in exact match accuracy (Figure 2) and confirmed that the bimodal random variation distribution persists when viewing a continuous performance metric (Figure 4). We can consider another continuous metric, which corresponds to the minimum probability assigned to a token for each sample in the test length generalization dataset:

$$\text{minprob}(f, X) = \frac{1}{|X|} \sum_{x \in X} \min_{i < |x|} \mathbf{P}_f[x_i | x_{0 \ldots i-1}] \tag{2}$$

We consider the average minimum probability (and analogously average maximum loss) in the sequence because errors are rare, and thus averaging across the sequence obscures generalization failures. However, as the model improves, the lowest-probability token may shift, but this transition is still continuous, not abrupt. Thus, unlike 0/1 accuracy, this metric also avoids threshold effects and better reflects gradual improvements in length generalization, similar to the continuous error metric. We plot the resulting histograms of this metric in Figures 15 and 16 for addition and count respectively. We note that bimodality persists for both tasks, particularly for addition; in particular, stronger bimodality exhibited in the accuracy histograms corresponds to stronger bimodality in the probability histograms.

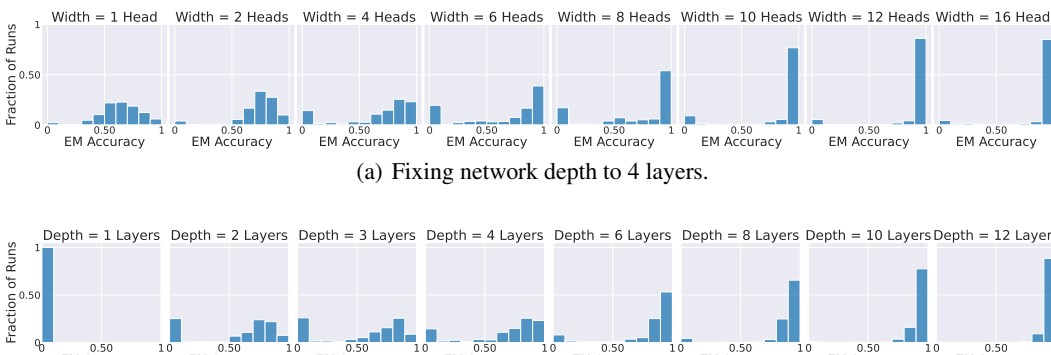

(a) Fixing network depth to 4 layers.

(b) Fixing network width to 4 heads per layer.

Figure 12: **Random variation in length generalization (count task).** Histograms of exact match accuracy on length 60 sequences when independently scaling (a) width and (b) depth.

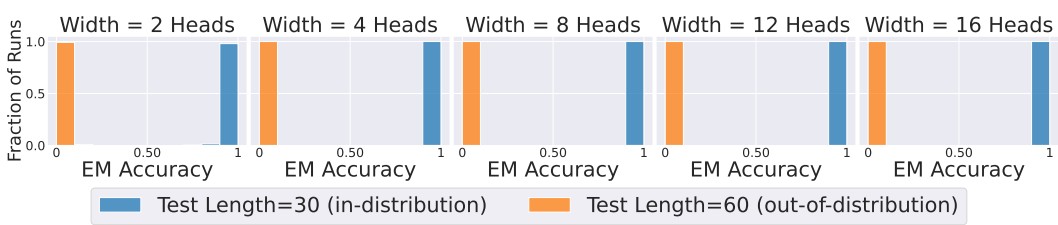

Figure 13: **Histograms for EM accuracy when fixing depth to be one layer (count task).** We plot the random variation distribution at test length 30 (blue, i.e. in-distribution) and test length 60 (orange, i.e. out-of-distribution). While all model seeds obtain near perfect accuracy in-distribution, all model seeds fail to length generalize at this depth.

In Figure 17, we plot the analogous KDE plot of the loss-based error metric for the count task. For count, we note that the mode corresponding to 'failing runs' when looking at accuracy or probability is more diffuse compared to addition. However, we claim that stronger bimodality in the original accuracy/probability plots in Figures 12 and 16 is still associated with stronger bimodality in the loss plots. For the addition task, the scales exhibiting the strongest bimodality in accuracy and probability are for width 4 and 8 (in the case of fixed depth) and depth 2 and 4 (in the case of fixed width), and these are also the most strongly bimodal loss KDE distributions. For the count task, even the most strongly bimodal setting (fixed width, with depth 2) has a low probability of failure ( 25%), but still produces a visible wide peak elevated over the long tail of the loss distribution.

Finally, Figure 14 provides a copy of Figure 8 with both accuracy and NLL loss comparable for easy contrast. We note that high-performance clusters are distinguishable in both accuracy and loss, but the thresholded accuracy conceals low-performance clusters which are visible in the continuous distribution.

# D ADDITIONAL LANGUAGE MODEL RESULTS

## D.1 MMLU RESULTS ON QWEN2.5-1.5B

We repeat the same MMLU experiment (Section 3.1) on Qwen2.5-1.5B and we include results in Figure 18. Due to computation limit, for each data mixes, we train on 20 random seeds. Similar to Figure 14. models yields the same bimodal distribution in accuracy metrics as well as multi-modal distribution in continuous metric (NLL).

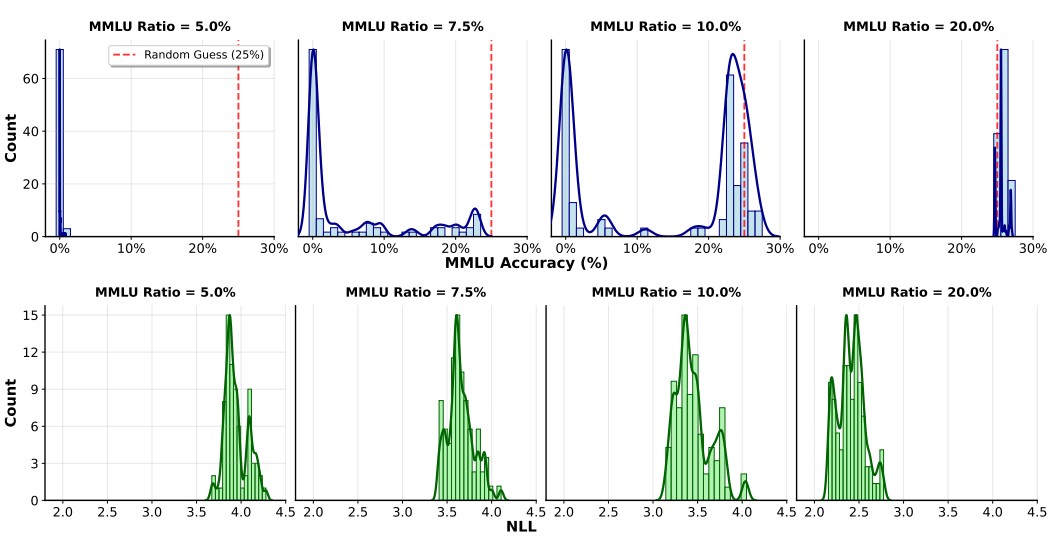

Figure 14: **KDE approximations of performance variation on the MMLU test set.** Provided for contrasting accuracy and continuous NLL loss. *Top:* Approximate distribution of accuracy metrics. *Bottom:* Approximate distribution of KDE metrics, copied from Figure 8.

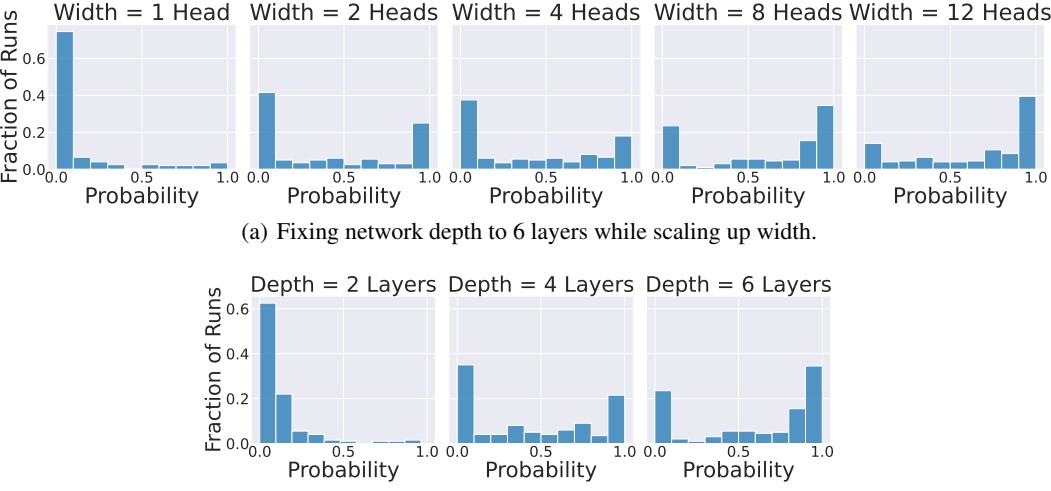

(a) Fixing network depth to 6 layers while scaling up width.

(b) Fixing network width to 8 heads per layer while scaling up depth.

Figure 15: **Histograms of the minimum probability of any token in each sequence, averaged across sequences (addition task).** Metrics are calculated by Equation 2. The bimodal nature of the random variation distribution persists even when using this continuous metric, analogous to Figure 4.

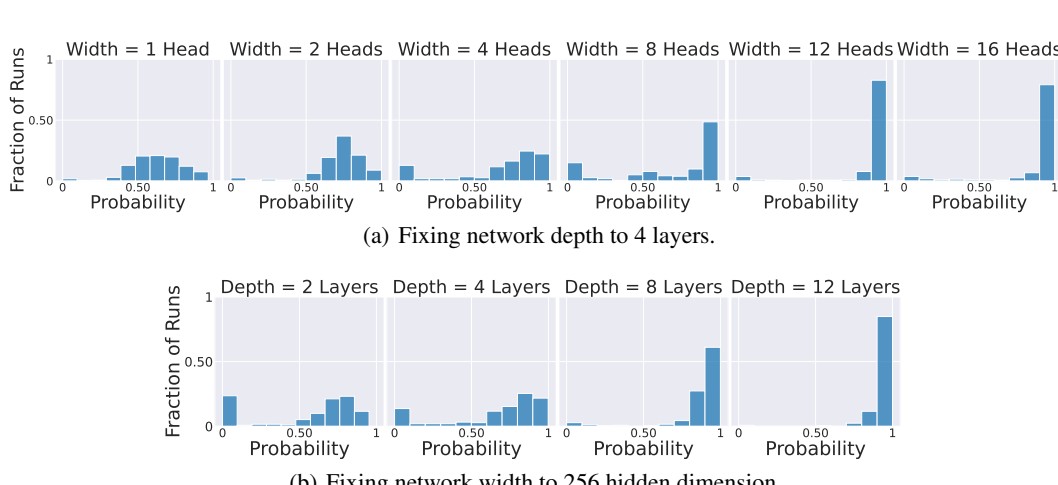

Figure 16: **Histograms of the minimum probability of any token in each sequence, averaged across sequences (count task).** Random variation still leads to bimodal performance distributions, even using this continuous performance metric.

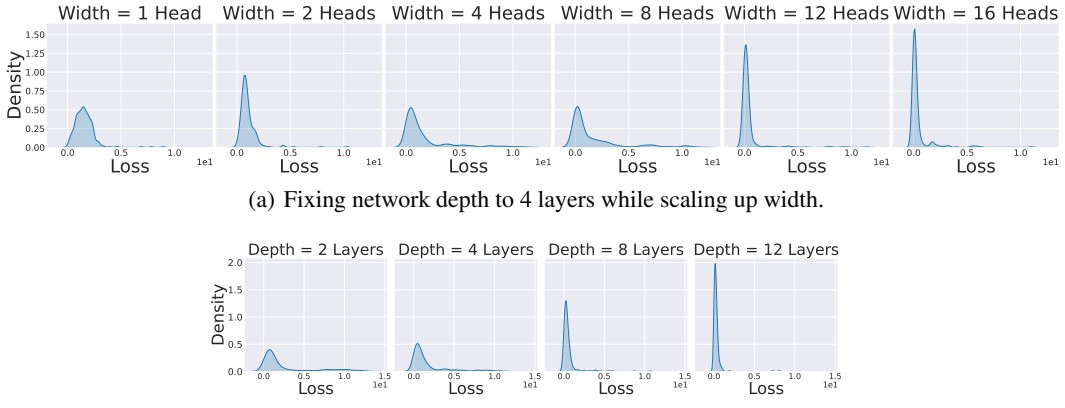

Figure 17: **Random variation in continuous length generalization error (count task).** Kernel Density Estimation (KDE) of loss-based error metric (Equation 1) distribution across model runs. At scales where the EM accuracy distribution is most strongly bimodal, we see that the KDE places the most density at areas of high loss.

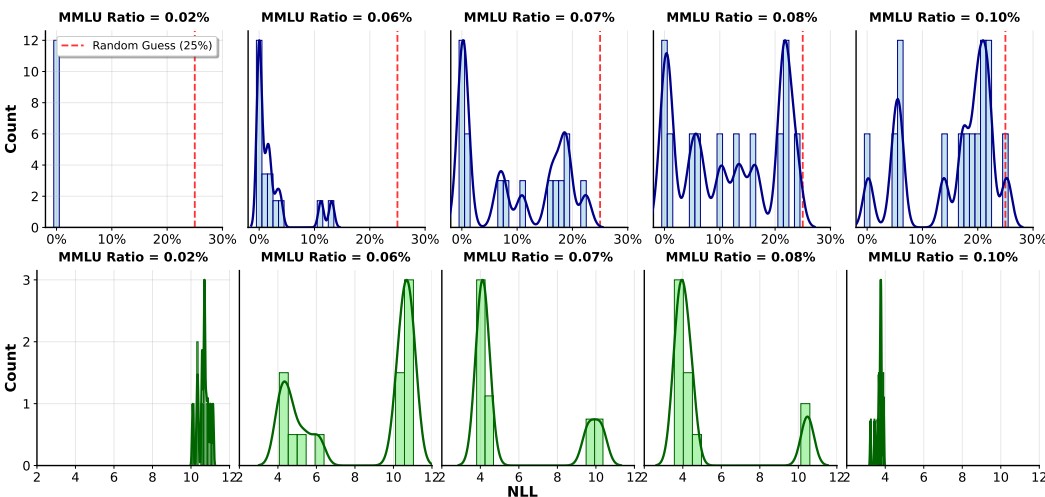

Figure 18: **Accuracy and NLL loss for Qwen2.5-1.5B MMLU experiments.** We train last-layer reinitialized Qwen2.5-1.5B on data mixes with different ratios of the target task (MMLU). *Top:* Distribution of accuracy metric. *Bottom:* Distribution of NLL loss metric.

## D.2   CSQA LM RESULTS

In addition to MMLU task, we also reproduce the same data mix experiment on CommonsenseQA (CSQA; Talmor et al., 2019) task and we include the result in Figure 19. Models yields the same qualitative behaviors as the MMLU task observed in Figure 6.

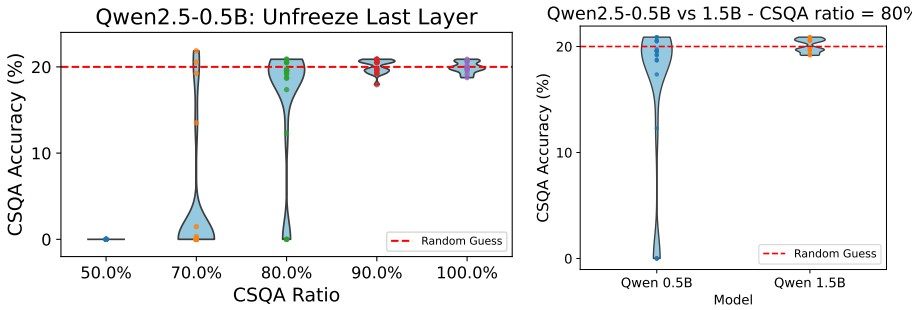

Figure 19:  Violin plots of last-layer reinitialized Qwen2.5 models on the CSQA benchmark. The same behavior observed in MMLU in Figure 6 is reproduced on CSQA as well.

