# OpenReview forum: "Random Scaling of Emergent Capabilities"
_ICLR.cc/2026/Conference — Submitted to ICLR 2026_

### Official Review · Reviewer_Fq9j · 2025-10-23

**Soundness:** 2
**Presentation:** 3
**Contribution:** 2
**Rating:** 4
**Confidence:** 3

**Summary:**

This paper presents the observation of the bimodal distribution of performance of models trained under different random seeds. Some seeds show emergent abilities, while some seeds display smooth improvement. The bimodal distribution can be used to explain some training dynamics. For example, the emergent abilities of LLM might be a manifestation where a smaller model falls into the left side of the distribution (bad performance) and a larger model falls into the right side (good performance). Also, the minimum capability can be identified when the performance distribution goes from unimodal (all on the left side) to bimodal. Experiments are conducted on two synthetic algorithmic tasks, MMLU, and CSQA.

**Strengths:**

1. The proposed observation of bimodal distribution is interesting and makes sense as a potential explanation for emergent abilities.
2. The emergence from unimodal to bimodal distribution as a sign of possessing minimum capability is an interesting and well-explained observation.
3. The paper is clear and easy to follow.

**Weaknesses:**

1. I would suggest changing the title of Section 2 from "Experiment" to "Experimental Setup." You only introduce the setup there.
2. Typos in lines 246-247: "we see that the probability (Figure 3.2 (bottom left) and mean (bottom right) of such “successful”." Throughout the paper, you seem to regard Figures 3 and 6, which have 4 subfigures, as being displayed as a 2*2 layout.
3. In lines 359 & 368, Figure 3.5 is mislinked to Figure 6.
4. In line 414, incorrect citation format. ("...process the multiple-choice format Hu and Frank (2024).")
5. Results of the synthetic task in Section 3 may not transfer well to benchmark datasets. Specifically, the experiments of MMLU and CSQA do not convince me:
    (1) The emergent abilities of MMLU happen at the emergence threshold, where model performance rapidly increases from 25% acc to 40% or so, as displayed in Figure 1 of [a]. In contrast, your experiments are on small models that are before the emergence threshold.
    (2) Figures 8 and 15 do not show a clear bi- or multi-modal distribution.

My understanding is that a slight perturbation on model weights (through different randomizations) affects its performance, leading to a steeper/flatter performance gap between two adjacent models. However, the effect of perturbation is eliminated as the model size grows (trained on more samples).

6. I would suggest that authors discuss this work's limitations in an independent paragraph or section.

a. [U-shaped and Inverted-U Scaling behind Emergent Abilities of Large Language Models](https://openreview.net/forum?id=jjfve2gIXe)

My main concern is 5.; MMLU and CSQA do not exhibit clear bimodal distribution in my opinion.

**Questions:**

1. [a] seems relevant to some of your arguments, such as the observation in Section 3.5 ("Competing solutions can lead to either monotonic or U-shaped trends in emergence likelihood."). They found that emergent abilities can be decomposed into a U-shaped trend and a double descent trend, which cancel out each other before the emergence threshold.

2. Can you explicitly state the formulas of overall mode, overall mean, success probability, and mean of successful runs in Figures 3 and 6?

3. Do you have any hypotheses for the cause of bimodal distribution?

4. Some works [a, b] argue the predictability of emergent abilities. Does this observation provide any new insights to help answer this long-lasting debate?

a. [U-shaped and Inverted-U Scaling behind Emergent Abilities of Large Language Models](https://openreview.net/forum?id=jjfve2gIXe)

b. [Predicting Emergent Capabilities by Finetuning](https://arxiv.org/abs/2411.16035)

---

> ### Author Response · Authors · 2025-11-25
> **Response (1/2)**
>
> Thank you reviewer Fp9j for your thoughtful response. We provide detailed comments below:
>
> > W1-4
>
> Thank you for identifying these typos, we have fixed the presentation issues in the updated PDF.
>
> > W5: Results of the synthetic task in Section 3 may not transfer well to benchmark datasets. Specifically, the experiments of MMLU and CSQA do not convince me: (1) The emergent abilities of MMLU happen at the emergence threshold, where model performance rapidly increases from 25% acc to 40% or so, as displayed in Figure 1 of [a]. In contrast, your experiments are on small models that are before the emergence threshold. (2) Figures 8 and 15 do not show a clear bi- or multi-modal distribution.
> My understanding is that a slight perturbation on model weights (through different randomizations) affects its performance, leading to a steeper/flatter performance gap between two adjacent models. However, the effect of perturbation is eliminated as the model size grows (trained on more samples).
>
> We appreciate the reviewer’s observation. Our experiments use the Qwen model family, where the 0.5B model already achieves around 40% accuracy on MMLU, corresponding to the emergence threshold reported in [a]. The apparent difference in the threshold scale arises because the training pipelines differ: the Qwen models are trained with higher-quality data and likely more tokens compared to those used in [a]. As a result, emergence for Qwen occurs at a smaller scale.
>
> (1) To quantitatively assess multimodality, we applied Hartigan’s Dip Test [1] to the distribution of MMLU scores across 80 random seeds. The Dip Test measures the maximum deviation (“dip”) between the empirical distribution and the best-fitting unimodal distribution. It then evaluates whether this deviation is large enough to reject the null hypothesis of unimodality. Intuitively, the resulting p-value reflects how likely the observed data could arise from a unimodal distribution. Concretely, lower p-values indicate stronger evidence for multimodality. As shown below, at the smallest data ratios (5%), the test does not reject unimodality. As we increase the data ratio, bi-modality emerges, the p-value at 7.5% is 0.11, already indicates the distribution is less likely to be unimodal. At 10% and 20%, the p-value drops below 0.001, strongly rejecting unimodality and confirming the emergence of a bimodal structure. Thank you for making the suggestion and we have updated the PDF to include this result.
> | MMLU Ratio | Dip  ↑ | p-value    |
> |-------------|-------|------------|
> | 5.00%       | 0.047 | 0.368 |
> | 7.50%       | 0.057 | 0.110  |
> | 10.00%      | 0.144 | < 0.001    |
> | 20.00%      | 0.107 | < 0.001    |
>
> (2) Thank you for providing this intuition. We agree that increased scale should reduce the impact of perturbations—indeed, a central point of our paper is that what looks like a “breakthrough” at a given scale is better understood as a shift in the underlying distribution over training outcomes as scale increases. In our setting, continuing to scale up the model should make the “failure” component increasingly low probability, so we do not necessarily see this as a weakness of the phenomenon we study, but rather as part of its explanation. To our knowledge, [2] shows that finetuning can distort pretrained features and harm OOD performance, but does not state that this effect can be straightforwardly mitigated purely by scaling model or data. If there are specific works supporting this claim, we would be very grateful for pointers so that we can more clearly connect our results to that literature.
>
>
> [1] The Dip Test of Unimodality (Ann. Statist. 13 (1) 70 - 84, March, 1985. https://doi.org/10.1214/aos/1176346577)
>
> [2] Fine-Tuning can Distort Pretrained Features and Underperform Out-of-Distribution https://arxiv.org/pdf/2202.10054
>
> > W6: I would suggest that authors discuss this work's limitations in an independent paragraph or section.
>
> We appreciate the suggestion and in the updated pdf we have now included a limitations section.

---

> > ### Author Response · Authors · 2025-11-25
> > **Response (2/2)**
> >
> > > Q2: Can you explicitly state the formulas of overall mode, overall mean, success probability, and mean of successful runs in Figures 3 and 6?
> >
> > At each model configuration (e.g., a fixed width/depth or data-mix setting), we have $N$ independent runs with exact-match accuracies $a_1, a_2, \dots, a_N \in [0,1]$. All four quantities in Figures 3 and 6 are computed from this empirical distribution $\\{a_i\\}_{i=1}^N$.
> > * Overall mean: Computed as the average accuracy $\text{OverallMean} = \frac{1}{N}\sum_{i=1}^{N} a_i.$
> > * Overall mode: For the mode, we first round each accuracy to the nearest fifth of a percent (i.e., increments of 0.05) obtaining $\\{\tilde{a_i}\\}_{i=1}^N$ and record the most frequently occurring value in this set.
> > * Success probability: Fix a success threshold \(\tau\) (e.g., \(\tau = 0.2\) in Figure 3). Define the success indicator $s_i = \mathbf{1}[a_i \ge \tau].$ Then the success probability is $\text{SuccessProb} =\frac{1}{N}\sum_{i=1}^{N} s_i = \mathbb{P}(a_i \ge \tau)$.
> > * Mean of successful runs: Conditioning on success ($s_i = 1$), the mean accuracy of successful runs is $\text{MeanSuccessful} =
> > \frac{\sum_{i=1}^{N} s_i\, a_i}{\sum_{i=1}^{N} s_i} \quad\text{(defined when } \sum_{i=1}^{N} s_i > 0\text{).}$
> >
> > > Q3: Do you have any hypotheses for the cause of bimodal distribution?
> >
> > This is an excellent question, and we agree that formally characterizing the origin of bimodality is an exciting direction for follow-up work. Our current hypothesis draws on the lottery ticket perspective: to exhibit a given capability, the model must form a specific circuit or subnetwork. Each attention head can be viewed as a potential candidate for forming such a circuit. However, if there are two possible circuits with similar train loss but different OOD behavior (generalizing vs non generalizing), an efficiently compressed model will only retain one of these circuits. In a sense, these different circuits are competing for which one gets the “best ticket”. If the generalizing circuit requires more parameters, then at small scales, the non-generalizing circuit’s “best ticket” is likely to outcompete the generalizing circuit’s “best ticket”, so the optimizer will focus on the non-generalizing circuit. As an example, see the count task: It requires at least a two layer model to generalize, whereas a non-generalizing circuit can achieve perfect in-distribution accuracy with only one layer.
> >
> > Through their effects on initialization and data ordering, random seeds determine which heads “win the lottery” and converge to the correct functional configuration. As model scale increases, the number of potential “tickets” grows, but the rate at which the expected “winnings” grow may be different for larger or smaller circuits, with larger circuits benefiting disproportionately from scale. Therefore, as scale increases, the “winning ticket” for a generalizing circuit is more likely to outperform the “winning ticket” among non-generalizing circuits, increasing the fraction of successful runs. This mechanism naturally explains the observed bimodality and its smooth shift with scale.
> >
> > > Q4: Some works [a, b] argue the predictability of emergent abilities. Does this observation provide any new insights to help answer this long-lasting debate?
> >
> > We thank the reviewer for raising this important connection. The cited works [a, b] focus on the predictability of emergent abilities, showing that performance trends can often be anticipated given sufficient information about pretraining scale or finetuning dynamics. Our findings complement this line of work by revealing why such predictability can appear inconsistent across experiments/studies. Specifically, we show that model performance near emergent thresholds can follow a bimodal distribution across random seeds, reflecting stochastic differences in whether a capability is acquired. From this distributional view, what appears to be a sudden, unpredictable “breakthrough” in a single training run is actually the surface manifestation of a continuous change in the probability of success across scales.
> >
> > Our results therefore bridge the two sides of the debate: emergent abilities are stochastically predictable when viewed as probabilistic transitions rather than deterministic thresholds. (We do also study how deterministic thresholds are still possible when models reach a specific capacity, clarifying that there are two different types of emergence, though the cases of interest are likely in the middle of the distribution shift we focus on.) In this sense, our work provides a unifying explanation—emergence is neither purely abrupt nor purely smooth, but arises from continuous, probabilistic changes that yield bimodal outcomes at the population level.
> >
> > Once again, we sincerely thank you for your time and thoughtful feedback, and we would be very grateful to hear about any remaining feedback or concerns that you feel we have not fully addressed.

---

### Official Review · Reviewer_Tnbg · 2025-10-30

**Soundness:** 2
**Presentation:** 2
**Contribution:** 2
**Rating:** 2
**Confidence:** 3

**Summary:**

Instead of studying emergent capabilities using a single training run or average of a few runs, this paper studies emergent capabilities using ~200 training runs of different random seeds. The authors attributed that "breakthroughs are instead driven by continuous changes in the probability distribution of training outcomes when performance is bimodally distributed across random seeds".

**Strengths:**

The paper studies emergent capabilities from a novel perspective, from a distributional perspective of many training runs instead of a single training run. This is important and helpful because neural network learning is inherently stochastic.

**Weaknesses:**

1. I could not agree with the paper's explanation that emergent capabilities are driven by the binomial distribution in capabilities, that "This variability is precisely what causes some model runs to appear as breakthroughs while others follow a more linear progression." I believe the causality should be the other way around. Some training runs show breakthrough, so that the capability performance improves abruptly from one mode to another. And other training runs show linear improvement. When these two kinds of training combine, they give the multi-modal distribution shown in Figure 2.

For example, thresholding is a mechanism that gives rise to discontinuity and that might cause emergent capability. I understand this as a valid cause driving emergent capability. But I believe how the distribution of capability performance changes during learning is more of a result/consequence of linear/emergent learning, rather than as cause/driving factor.

I acknowledge that the distribution of different training runs, rather than one single training run, is worth study regarding emergent capability. My concerns are regarding the causality.

2. The authors conduct experiments using reinitializations rather than training from scratch for computational cost constraints. Reintializations involves reinitialize the final attention layer and the subsequent LM head, while keeping most other layers as trained. I believe this experimental setup differs significantly from a from-scratch training, and might change the learning behaviour. Could the authors provide evidence that support the eligibility of such approach for studying emergent capabilities? for example, are there other works on studying emergent capabilities that use similar reinitialisation instead of training from scratch?

3. The Section 4.3 explains Figure 8 as roughly bi-modal. I think it is ambiguous from reading the figure alone. It also looks reasonable to me that the MMLU ratio = 7.5% figure and the MMLU ratio = 20% figure are unimodal. Could the authors more rigorously use a standard to test whether they are bimodal, for example statistical tests?

4. Finally, I feel the paper is quite dense in terms of experiments and explanations and I feel personally challenging to grasp the main take-away. The authors are encouraged to improve the ease of reading.

**Questions:**

See above

---

> ### Author Response · Authors · 2025-11-25
> **Response (1/2)**
>
> Thank you reviewer Tnbg for your thoughtful response. We provide detailed comments below:
>
> > I could not agree with the paper's explanation that emergent capabilities are driven by the binomial distribution in capabilities, that "This variability is precisely what causes some model runs to appear as breakthroughs while others follow a more linear progression." I believe the causality should be the other way around.
>
> Thank you for raising this point regarding causality. While we agree that it might make sense in some cases to say “scaling breakthroughs” cause “bimodal variation”, we believe that what is labeled as “emergence with scale” is, in many cases, better understood as a change in the mixture over seeds. Importantly, we notice two types of scale emergence.
>
> In one type, we have a gradual-shift regime (which we observe for real-world tasks like multiple-choice QA) where individual runs already cluster into “success” and “failure” modes at smaller scales; as scale increases, the probability of landing in the successful mode increases smoothly. In the other type, we have  threshold-based regime (which we see more clearly in the synthetic counting task), capacity constraints do induce a more genuine phase transition: the bimodality itself only appears once the model is large enough to implement the underlying algorithm, at which point success probability jumps sharply. In our counting task, this distinction is particularly visible as the success probability jumps from 0% to 85% when we add a layer.
>
> In discussions of emergence, it is common for researchers to point to apparent “magic numbers” in parameter count at which capabilities suddenly appear, such as the “multiple-choice onset.” Our results suggest that while some tasks do exhibit genuine capacity thresholds of this kind, many reported emergence points are better explained by stochastic variation across seeds rather than a strict critical scale. When one examines the unstable transition region more closely, the underlying success probabilities often change quite gradually—much more slowly than the abrupt jumps suggested by any single sampled scaling curve.
>
> In terms of implications, we think our findings should reshape how emergence and random variation are discussed, but a more practical outcome is that scale curves should be accompanied by statistical tests that check whether deviations from smooth trends are genuinely irregular. As long as the community remains interested in unusual scaling behavior—and in exploiting data or training strategies to “beat” standard scaling laws—evaluations will continue to surface seemingly sudden breakthroughs. Our results suggest that, in these cases, we should explicitly ask: how likely is this jump to arise when sampling from an underlying unimodal or bimodal distribution?
>
> > The authors conduct experiments using reinitializations rather than training from scratch for computational cost constraints. Could the authors provide evidence that support the eligibility of such approach for studying emergent capabilities? for example, are there other works on studying emergent capabilities that use similar reinitialisation instead of training from scratch?
>
> Fine-tuning only the LM head, or reinitializing and training a new head on top of a pretrained backbone, is a standard practice in language model adaptation. For instance, in the Linear Mode Connectivity literature (e.g., [1, 2]), one keeps the backbone fixed and trains only the output head, then studies how the resulting solutions are connected within the same loss basin. This line of work supports the view that such reinitialization-based protocols can be informative about model behaviour and capabilities, even without retraining the entire network from scratch.
>
> We agree that fine-tuning can, in principle, amplify differences between runs by distorting or reweighting existing features [3]. However, for such amplification to occur, the underlying differences must already be present in the full-rank loss surface of the pretrained model family; the head can only exploit or accentuate variation that is already “available” in the backbone representations. In this sense, our protocol still probes genuine structure in the training dynamics. Moreover, this setting is directly relevant to post-training practice, where it is well documented that variation across runs (e.g., in RLHF or instruction tuning) can be substantial. Our results therefore speak not only to the underlying loss landscape, but also to the practical reality that post-training procedures can expose and magnify seed-level differences that are already implicit in the model.
>
> [1] Linear Connectivity Reveals Generalization Strategies https://arxiv.org/pdf/2205.12411
>
> [2] Exploring Mode Connectivity for Pre-trained Language Models https://arxiv.org/pdf/2210.14102
>
> [3] Fine-Tuning can Distort Pretrained Features and Underperform Out-of-Distribution https://arxiv.org/pdf/2202.10054

---

> > ### Author Response · Authors · 2025-11-25
> > **Response (2/2)**
> >
> > > The Section 4.3 explains Figure 8 as roughly bi-modal. I think it is ambiguous from reading the figure alone. It also looks reasonable to me that the MMLU ratio = 7.5% figure and the MMLU ratio = 20% figure are unimodal. Could the authors more rigorously use a standard to test whether they are bimodal, for example statistical tests?
> >
> > To quantitatively assess multimodality, we applied Hartigan’s Dip Test [1] to the distribution of MMLU scores across 80 random seeds. The Dip Test measures the maximum deviation (“dip”) between the empirical distribution and the best-fitting unimodal distribution. It then evaluates whether this deviation is large enough to reject the null hypothesis of unimodality. Intuitively, the resulting p-value reflects how likely the observed data could arise from a unimodal distribution. Concretely, lower p-values indicate stronger evidence for multimodality. As shown below, at the smallest data ratios (5%), the test does not reject unimodality. As we increase the data ratio, bi-modality emerges, the p-value at 7.5% is 0.11, already indicating the distribution is less likely to be unimodal. At 10% and 20%, the p-value drops below 0.001, strongly rejecting unimodality and confirming the emergence of a bimodal structure. Thank you for making the suggestion and we have updated the PDF to include this result.
> > | MMLU Ratio | Dip  ↑ | p-value    |
> > |-------------|-------|------------|
> > | 5.00%       | 0.047 | 0.368 |
> > | 7.50%       | 0.057 | 0.110  |
> > | 10.00%      | 0.144 | < 0.001    |
> > | 20.00%      | 0.107 | < 0.001    |
> >
> > > Finally, I feel the paper is quite dense in terms of experiments and explanations and I feel personally challenging to grasp the main take-away.
> >
> > We appreciate this feedback and agree that the original presentation was dense, especially in the synthetic experiments. To clarify the main takeaway, we have revised the manuscript to have a more focused narrative for the synthetic experimental results. In particular, Section 2 now focuses on the reverse-order addition task as a representative example that illustrates our key phenomenon: seemingly bimodal “random” variation in performance across runs actually reflects gradual shifts in the underlying distribution, and this structure is also visible in continuous metrics, not just exact-match accuracy. For ease of presentation we have moved results on Count and discussion on the inverse-scaling-related observations to Appendix C.1.
> >
> > Once again, we sincerely thank you for your time and thoughtful feedback, and we would be very grateful to hear about any remaining feedback or concerns that you feel we have not fully addressed.

---

> > > ### Comment · Reviewer_Tnbg · 2025-11-27
> > >
> > > The authors have answered satisfactorily to some of my questions, but I still have some questions regarding Section 3.4, Figure 8, and Table 1.
> > >
> > > 1.
> > > In line 437, "In Figure 8, we find that continuous negative log likelihood (NLL) loss metrics remain highly multimodal."
> > > But in line 449, "Therefore, the continuous metrics remain roughly bimodal".
> > > I wonder, and it would be helpful for the authors to make it clearer in main text in Section 3.4, whether they are arguing the continuous NLL metric to be multi-modal or bimodal. If multi-modal, there seems to be a missing link between multi-modal distribution in continuous metrics and bimodal distribution in performance.
> > >
> > > 2. From Table 1, I think it can only be said that MMLU Ratio 10% and 20.00% are not unimodal, the other two entries MMLU Ratio 5% and 7.5% are statistically weak to say anything meaningful.
> > > But the authors said in line 449 that "Therefore, the continuous metrics remain roughly bimodal—as is especially clear in the 10% mix setting". I think it is not supported by Table 1 and Figure 8 because multimodal might not necessarily be bimodal.
> > >
> > > 3. Visually from Figure 8, MMLU Ratio 5% does looks like bimodal because it has two apparent peaks but Table 1 cannot reject it being unimodal.
> > >
> > > So in summary, I think there is some inconsistency regarding Section 3.4, Figure 8, and Table 1. The authors are encouraged to make it clearer in the main text.

---

### Official Review · Reviewer_JeTL · 2025-10-31

**Soundness:** 3
**Presentation:** 3
**Contribution:** 2
**Rating:** 4
**Confidence:** 2

**Summary:**

This paper studies "emergent" capabilities--sharp breakthroughs/increases in model performance at a sufficiently large model size--and provides empirical support for the view that model performance across scales is bimodally distributed across random seeds. Through experiments on both synthetic and natural language tasks (MMLU), they show that some seeds exhibit linear trends, while others show emergent trends; they argue that when reporting results on individual seeds, results are likely skewed to the modes of the underlying distributions of model performances at scales. They find that these results hold for both discrete and continuous metrics (loss).

**Strengths:**

- The paper provides a careful conceptual lens through which to view the well-studied phenomenon of emergent capabilities. This viewpoint is empirically well-supported.
- Experiments are extensive, showing the robustness of results for a wide range of metrics (continuous vs discrete, mode vs mean) across seeds and datasets (synthetic and real-world).

**Weaknesses:**

- Potential for impact: Although the finding that not all individual seeds themselves exhibit non-linearity in emergent capabilities is interesting, it is not clear what the impact of the empirical findings in the work are. If what appears as emergence is that the mode of the performance distributions sharply increases, is this not a form of emergence? What are the implications of this work for how we study and evaluate models?
- Some analysis decisions are arbitrary: For example, why is 20% exact match accuracy used as the threshold for success in Section 3.2?
- Limited models: All analyses on real-world tasks are with Qwen models, not models from other families.

**Questions:**

Minor Notes:
- Figure 3 is referred to as Figure 3.2 in the paper, but there is only a single figure. Line 247: It is also unclear to me what "bottom left" and "bottom right" mean here.
- Line 247: What do depths 2 and 3 correspond to in the figure?

---

> ### Author Response · Authors · 2025-11-25
>
> Thank you reviewer JeTL for your thoughtful response. We provide detailed comments below:
> > Potential for impact: Although the finding that not all individual seeds themselves exhibit non-linearity in emergent capabilities is interesting, it is not clear what the impact of the empirical findings in the work are. If what appears as emergence is that the mode of the performance distributions sharply increases, is this not a form of emergence? What are the implications of this work for how we study and evaluate models?
>
> We appreciate this question regarding impact and interpretation. Our work directly extends the discussion from Schaeffer et al. [1], who observed that only 8% of reported breakthroughs persisted under continuous metrics, and hypothesized that these may simply reflect random noise. However, that work did not establish whether such “emergent” behaviors in continuous metrics are statistically significant. Our contribution is to resolve this ambiguity by reframing emergence as a distributional phenomenon: when performance is measured across random seeds, apparent breakthroughs correspond to the onset of bimodality in the performance distribution. This perspective explains why continuous metrics can still exhibit sharp transitions. Namely, these reflect a shift in the probability mass of successful runs rather than mere noise. Conceptually, our framework lays the groundwork for a distributional view of emergence, allowing future studies to determine whether observed breakthroughs are statistically significant or artifacts of stochastic variation.
>
> > Some analysis decisions are arbitrary: For example, why is 20% exact match accuracy used as the threshold for success in Section 3.2?
>
> The choice of 20% exact-match accuracy as the success threshold is not specific to our conclusions; any value in the “valley” between the failing and successful modes would yield qualitatively similar trends. As shown in Figure 2, the distribution of accuracies is bimodal, with one mode near failure and another at substantially higher accuracy. We select 20% simply as a representative point between these modes that (i) lies in the low-density region separating failure and success, and (ii) yields sufficient dynamic range so that both the success probability and the mean-of-successful-runs curves have non-trivial variation across scales. We find our results are robust to reasonable changes in this threshold.
>
> > Limited models: All analyses on real-world tasks are with Qwen models, not models from other families.
>
> The Qwen models we study use a standard Transformer architecture, so we expect similar distributional trends to hold for other model families such as LLaMA or OLMo. Qwen already exhibits these behaviors at comparatively small scales (0.5B–1.5B parameters), whereas running an equivalent 80-seed, multi–data-ratio study for other architectures at these scales would be beyond our current compute budget.
>
> > Minor notes (presentation)
>
> We have fixed all the presentations in the updated PDF. To answer your question,  depths 2 and 3  in Line 247 correspond to the network depth (i.e., number of attention layers).
>
> [1] Are Emergent Abilities of Large Language Models a Mirage? (https://arxiv.org/abs/2304.15004)
>
> Once again, we sincerely thank you for your time and thoughtful feedback, and we would be very grateful to hear about any remaining feedback or concerns that you feel we have not fully addressed.

---

### Meta-Review · Area_Chair_CVSW · 2025-12-09

**Summary:**

This study investigates emergent behaviors w.r.t. model scale in language models. Across many random seeds, it is observed that model performance is distributed bimodally: some seeds exhibit linear trends w.r.t. scale, while others exhibit more emergence-esque trends. This is attributed to the mode of the models' performance for a given seed, rather than discrete vs. continuous metrics as in past work.

This batch of reviews varied in their concerns. Most prominently, Reviewer Tnbg questioned the direction of causality in the observed phenomena, and Reviewer JeTL questioned the impact of this line of thinking. While seemingly distinct concerns, these may be conceptually related: they reflect a skepticism regarding whether this perspective resolves any true mystery underlying emergence in a way that would actually enable us to predict future instances of emergence or apply this understanding to practical problems. I am inclined the share this skepticism.

**Reviewer Concerns:**

The primary concerns raised above have not been fully addressed in the discussion. Some interesting points were raised therein, including that emergence will continue to arise while the field continues to be impressed by it for its own sake (regardless of whether this is merely a statistical artifact or reflects genuinely interesting qualitative differences in behavior with scale). However, the implications of the specific claims made in the paper remain fuzzy and of questionable practical importance.

Concerns regarding experimental design raised by Reviewer JeTL have been sufficiently addressed, in my opinion.

Concerns raised by Reviewer Fq9j were relatively minor, and many of these have also been sufficiently addressed.

**Reviewer Scores:**

I think that the discussions may have yielded a minor score increase from Reviewer Fq9j. I do not believe that the other reviewers would have been likely to change their scores significantly, but I am not confident in this assessment.

---

### Decision · Program_Chairs · 2026-01-26

Reject